# Water Photo-Oxidation over TiO₂—History and Reaction Mechanism

Yoshio Nosaka 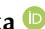

Department of Materials Science and Technology, Nagaoka University of Technology, 1603-1 Kamitomioka, Nagaoka 940-2188, Japan; nosaka@nagaokaut.ac.jp

**Abstract:** Photocatalytic water oxidation over titanium dioxide (TiO₂) was overviewed by surveying briefly the history of water photo-oxidation, followed by profiling the research for the molecular mechanism of oxygen evolution reaction (OER) at the TiO₂ surface. As the experimental approach to investigate the reaction mechanism, ESR, NMR, and STM were described as well as FTIR spectroscopy. Detection of reactive oxygen species, which are the intermediate species in the OER, was also involved in discussing the mechanism. As the theoretical approach to the reaction mechanism, some research with density functional theory (DFT) for anatase (101) surface was illustrated. Since the OER activity of rutile TiO₂ is higher than that of anatase, and the rutile (011) surface has been assigned to the oxidation facet, we performed a DFT calculation for a (011) surface model molecule. The results were successfully discussed with the reported mechanism. The first oxidation step occurs at the bridging OH site, which faces a Ti₅$_{5C}$ site. The water molecule which coordinates both sites is oxidized, and the resultant radical coordinates the Ti₅$_{5C}$ site to form a trapped hole Ti-O•. In the second step, a coordinated water molecule is oxidized at the Ti-O• site to form a Ti-OOH structure.

**Keywords:** photocatalysis; water splitting; titanium dioxide; oxygen evolution reaction; rutile; electron spin resonance; nuclear magnetic resonance; Fourier transfer infrared resonance; scanning tunneling microscope; density functional theory





## 1. Introduction

Photocatalytic splitting of water is a direct and attractive approach for the utilization of solar energy by producing the most-prospective clean hydrogen fuel. In photocatalytic water splitting, oxidation of water to molecular oxygen, or oxygen evolution reaction (OER), is the most difficult process because it needs the transfer of four electrons, while the hydrogen evolution reaction (HER) is a two-electron transfer reaction. Since these electron transfer reactions originated from the absorption of one photon for each step, the semiconducting properties of metal oxide solid materials are useful to promote the reaction at a reaction site [1]. Thus, metal complexes and metal–organic frameworks (MOF) are almost certainly unsuitable as photocatalysts for OER [2].

Photo-oxidation of water was reported a century ago with ZnO powder, and 50 years ago, TiO₂ was found to be useful as the stable material for converting photoenergy to chemical energy without deterioration. By starting TiO₂ as photocatalysts, this 50 years is a booming period in the research of photocatalysis. In the present report, the first section is devoted to the brief history of water photo-oxidation. In the following section, several research reports about the reaction mechanism for water photo-oxidation over TiO₂ are described. The reports for the reaction mechanism were based on either an experimental approach or a theoretical calculation approach. In the experimental approach, techniques used were electron spin resonance (ESR), nuclear magnetic resonance (NMR), scanning tunneling microscopy (STM), and Fourier transfer infrared spectroscopy (FTIR). In the theoretical calculation approach, several methods of density functional theory (DFT) have been used for anatase (101) and rutile (110) surfaces. In the following subsection, our recent results of DFT calculation for a model molecule of rutile (011) surface will be described. By

the present compilation, a common feature of the reaction mechanism for water oxidation at the $TiO_2$ surface will be clarified.

## 2. History of Photocatalytic Water Photo-Oxidation

After the first report of ZnO powder as the photosensitizer in 1911 [3], Baur and Perret reported the photoinduced oxidation of water to produce $O_2$ with ZnO powder in 1924 [4]. In this report, a new concept has been suggested, in which, after the absorption of light energy, the ZnO particle undergoes simultaneous anodic and cathodic processes. In the anodic process, $OH^-$ is oxidized to generate $1/4 O_2 + 1/2 H_2O$. On the other hand, in the cathodic process, the $Ag^+$ ion is reduced to generate Ag by the formation of intermediate peroxide $Ag_2O$. However, this photoinduced reaction is associated with the dissolution of ZnO to release $Zn^{2+}$, and then the verified reaction scheme was

$$2AgNO_3 + ZnO = 2Ag + Zn(NO_3)_2 + 1/2 O_2 \tag{1}$$

Therefore, though this reaction takes place only under photoirradiation, it was not a photocatalytic reaction in our time.

In the field of electrochemistry, a single crystal of ZnO was employed, and the photocurrent was observed under the high anodic polarization in water [5]. $O_2$ evolved with the anodic photocurrent. The evolution of $O_2$ was not attributable to the photo-oxidation of water but to the photo dissolution of ZnO, similar to the case of ZnO powder stated above. In this case, $Zn^{2+}$ was dissolved in the solution. Therefore, the total reaction is simply

$$2ZnO + h\nu \rightarrow 2Zn^{2+} + O_2 + 4e^- \tag{2}$$

The reaction mechanism is reported as follows. The holes photo-generated in the electrode oxidize the surface O of the electrode and the second hole attacks the neighbor surface O atom to form an $(O\text{-}O)^{2-}$ ion at the surface, then it becomes $O_2$ with the other two holes [5].

Compared with ZnO, $TiO_2$ is a relatively new material because titanium metal was identified at the end of the 18th century. It was in 1923 that $TiO_2$ was first manufactured as a white pigment to replace basic lead carbonate which was used in the porcelain industries. Photo-induced effects on the chemical reactions over $TiO_2$ powder have been investigated before and after World War II in the vicinity of the 1940s. However, as a photosensitizer, in other words, for photocatalytic reaction, the main substance used was ZnO since the photoeffect was first reported as described above. In 1953, Markham and Laidler [6] reported the reaction mechanism of photo-oxidation on the surface of ZnO, where they suggested that $O_2$ is formed from two •OH radicals. In the discussion, they cited the report of an XRD study [7], which described that photo-excitation of $TiO_2$ caused the change in the XRD pattern to $\alpha$-$Ti_2O_3$ crystal. Then, Markham and Laidler described that the absorption of photon energy to $TiO_2$ may result in the photolysis of the $TiO_2$ crystal, as represented by [6].

$$TiO_2 + h\nu \rightarrow 1/2 Ti_2O_3 + 1/2 O_2 \tag{3}$$

At that time, this chemical equation had not been confirmed by any researchers, and $TiO_2$ became inexperienced material in the research of solid photosensitizers.

About 50 years ago, in an electrochemical study, Fujishima and Honda used $TiO_2$ single crystals as photoelectrodes and examined photoinduced reactions. They observed the anodic photocurrent and $O_2$ evolution similarly to the case of ZnO. Surprisingly, in the case of the $TiO_2$ electrode, they could not detect $Ti^{4+}$ nor $Ti^{3+}$ ions in solution after the evolution of oxygen [8], which was different from the case of ZnO. This observation means the oxidation of water to generate $O_2$ with the aid of photo-energy as described by

$$TiO_2 + h\nu + 2H_2O \rightarrow O_2 + 4H^+ + 4e^- \tag{4}$$

They suggested that when a suitable *p*-type semiconductor electrode is coupled with the *n*-type $TiO_2$ semiconductor electrode, efficient electrochemical photolysis of water may occur on the irradiation of both electrodes [9]. Their report was epoch-making because it was shown that the photon energy could convert to the chemical energy of hydrogen fuel by using a semiconductor electrode. Actually, Yoneyama and co-workers demonstrated the $O_2$ evolution at the $TiO_2$ anode and the $H_2$ evolution at the *p*-GaP cathode with the open circuit voltage of 0.58 V [10]. Though the deterioration in the cell performance was observed due to the unstable *p*-GaP cathode, oxidation of water at the $TiO_2$ anode was confirmed stable. After these observations, the photoelectrochemical systems with metal oxides for water splitting have been investigated widely, which are reviewed by Rajeshwar [11]. Recent development in the engineering nanostructure interface of photoanode materials toward photoelectrochemical water oxidation can be referred to with a review article [12].

For the particulate photocatalysts, the decomposition of water vapor to $H_2$ and $O_2$ over Fe-doped $TiO_2$ powder has been reported by Schrauzer and Guth [13], in which a stoichiometric (2:1) evolution of $H_2$ and $O_2$ was confirmed. Photocatalytic water vapor decomposition was enhanced when Pt nanoparticles were photodeposited on a $TiO_2$ particle, which constructs a small particulate electrochemical cell. Sato and White [14] reported the generation of $O_2$ and $H_2$ with the stoichiometric ratio of 1:2, as shown in Figure 1. When the irradiation was stopped, the recombination of $O_2$ and $H_2$, or reverse reaction of $H_2O$ photodecomposition, occurred at a significant rate. Photocatalytic decomposition of water vapor was recently reviewed by Suguro et al. [15].

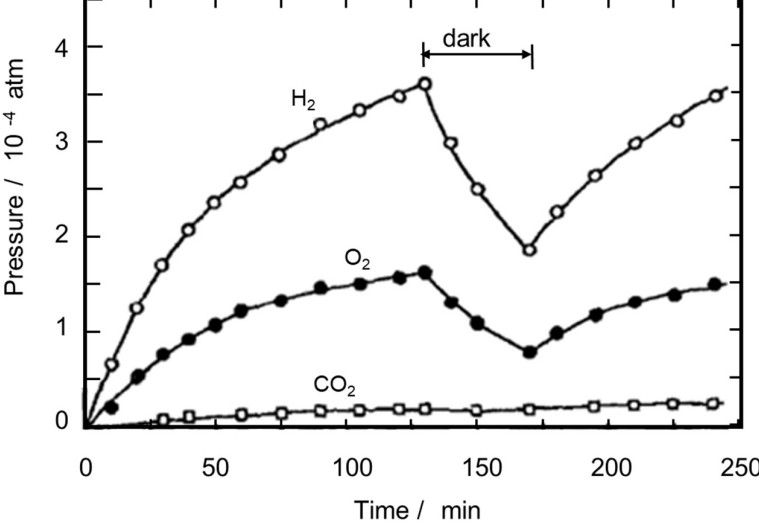

**Figure 1.** Photocatalytic water splitting over Pt-loaded $TiO_2$ powder in the gas phase. Reprinted with permission from Ref. [14]. Copyright 1980 Elsevier.

In the aqueous suspension system of $TiO_2$ powder, Sayama et al. [16] found that the stoichiometric $H_2$ and $O_2$ production was improved by adding carbonate salts to the suspension of the photocatalyst. Recently, they confirmed the mechanism of the improvement by the DFT calculation for model molecules [17]. The adsorbed $H_2CO_3$ molecule was photo-oxidized relatively easily compared with the adsorbed $H_2O$ molecule, facilitating the formation of the peroxide intermediate and improving both the $O_2$ evolution and the $H_2O_2$ production.

Different from the gaseous system, in the aqueous suspension system, it is difficult to inhibit the reverse reaction of the produced $H_2$ and $O_2$. To inhibit the reaction between $H_2$ and $O_2$, the surface of the co-catalyst for $H_2$ production had to be covered with a $Cr_2O_3$ shell to protect it from the attack of the produced $O_2$ [18]. From the point of view of a catalyst, $TiO_2$ is not the best material for OER. The other metal oxides which show a small exchange current for OER have been explored as reviewed by Lewis and coworkers [19]. Moreover, these materials have been employed as the co-catalyst of semiconductor photocatalysts.

For water oxidation, many co-catalysts, such as cobalt oxide, have been used in the newly developed photocatalysts [20]. The development of photocatalysts for water splitting of the use of such co-catalysts was compiled recently [21].

### 3. Mechanism for TiO$_2$ Water Oxidation

*3.1. Experimental Approach*

3.1.1. ESR and NMR

In the early days of photocatalysis, a spin-trapping ESR technique was used to detect radical species in the reaction. With the UV irradiation on the suspension of Pt-deposited TiO$_2$ powder, •OH and •O$_2$H radicals were detected [22]. Thus, in the O$_2$ generation by the water oxidation, •OH radical was considered as the reaction intermediate as follows.

$$H_2O + h^+ \rightarrow \bullet OH + H^+ \tag{5}$$

$$2\bullet OH \rightarrow H_2O_2 \tag{6}$$

$$2H_2O_2 \rightarrow O_2 + 2H_2O \tag{7}$$

The •O$_2$H radical detected was attributed to being produced by the photocatalytic reduction of the produced O$_2$ [22].

In the recent study, one- and two-dimensional $^1$H solid-state NMR techniques were employed to identify the surface hydroxyl groups and the adsorbed water molecules as well as their spatial proximity/interaction in TiO$_2$ photocatalysts [23]. Only the bridging OH (i.e., OH$_{br}$) is in close spatial proximity to adsorbed H$_2$O, forming hydrated OH$_{br}$. To investigate the role of hydrated OH$_{br}$ in the hole transfer process, in situ ESR experiments were performed on TiO$_2$ with variable H$_2$O loading [23]. The ESR measurements revealed that the hydrated OH$_{br}$ groups offer a channel for the transfer of photogenerated holes in the photocatalytic reaction, and the adsorbed H$_2$O could have a synergistic effect with the neighboring OH$_{br}$ groups to facilitate the formation and evolution of active paramagnetic intermediates. On the basis of these experimental observations, the detailed photocatalytic mechanism of water splitting on the surface of TiO$_2$ was proposed, as shown in Figure 2 [23]. The surface-trapped hole Ti-O• and the adsorbed •O$_2$$^-$ radical were identified by the ESR measurements. In this figure, as the product of the second step, they suggested the side-on coordination of O$_2$$^{2-}$ ions to one Ti ion. However, this structure is probably unreal conformation in an aqueous solution.

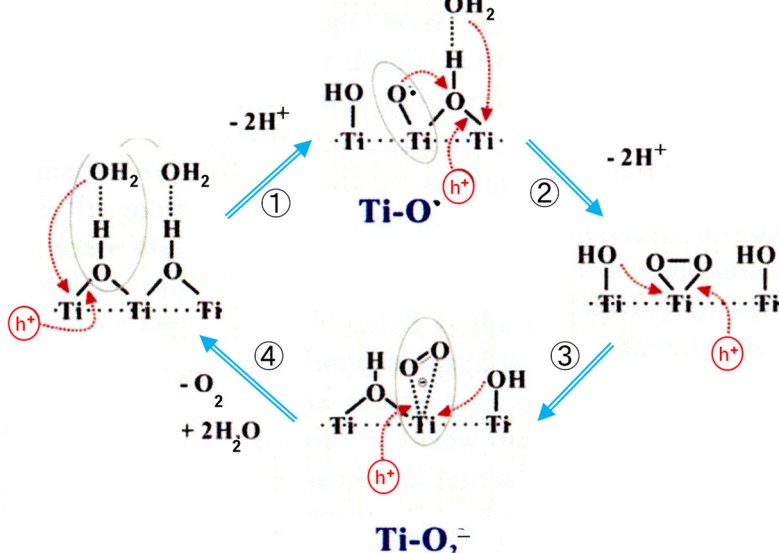

**Figure 2.** OER mechanism proposed from the ESR detection of radicals produced on Pt/TiO$_2$ (P25) powder. Adapted with permission from [23]. Copyright 2017, American Chemical Society.

### 3.1.2. STM

By using the low-temperature STM performed at 80 K, the dissociation of individually adsorbed water molecules has been observed at the five-fold coordinated Ti ($Ti_{5c}$) sites of the reduced rutile $TiO_2$(110)-1 × 1 surface under the irradiation of UV lights with the wavelength shorter than 400 nm [24]. It was found that two kinds of hydroxyl species are involved in the process of photocatalytic water dissociation. One is always present at the adjacent bridging oxygen sites, $OH_{br}$, and the other either occurs as terminal $OH_t$ at $Ti_{5c}$ sites away from the original ones or even desorbs from the surface [24]. This observation is in harmony with the suggestion of the solid-state NMR described above. Thus, the initial oxidation step could be expressed by the following equation [25].

$$H_2O_{Ti} + O_{br} + h^+ \rightarrow \bullet OH(gas) + OH_{br} \tag{8}$$

Using a combination of STM and other surface science techniques, the recent progress that provides fundamental insights into $TiO_2$ photocatalysis were reviewed through direct tracking of the evolution in single molecule photochemistry on $TiO_2$ single crystal surfaces [25].

### 3.1.3. FTIR

Nakato and co-workers investigated the molecular mechanism of water photo-oxidation reaction at atomically flat $n$-$TiO_2$ (rutile) single crystal photoelectrodes of (100) and (110) facet surfaces [26]. They measured the photoluminescence (PL) of the photoelectrodes in aqueous solutions of various pH values. Combining with the Fourier transfer infrared (FTIR) observations of the intermediates in the reduction of molecular oxygen as well as the oxidation of water with an electron acceptor of $Fe^{3+}$ ions, the oxidation process was elucidated, as shown in Figure 3. Though the nucleophilic attack of water to Ti−O$^+$−Ti is associated with the [Ti−O• HO−Ti] formation in Figure 3 [27], an alternative model was proposed in which the bridging oxygen radicals [Ti−OH• Ti] are photogenerated with the intrinsic band-gap surface state [28]. In both reaction models, the surface-trapped holes become bridging peroxo (Ti-OO-Ti) by combining with the hole generated secondarily in the same crystallite [29,30]. The bridging peroxo structure was regarded as the intermediate step of water photo-oxidation. Thus, the species generated by two-hole oxidation is equivalent to a chemically adsorbed $H_2O_2$.

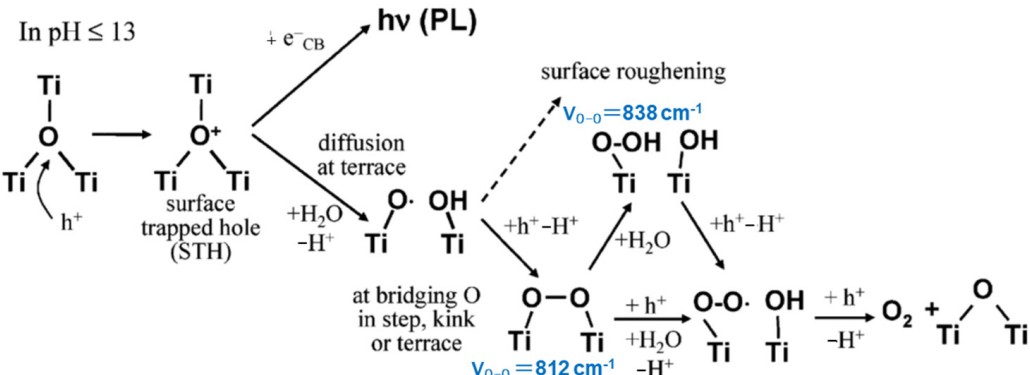

**Figure 3.** The proposed process of water oxidation and the peak position of the IR–spectra for the intermediates. Adapted with permission from Ref. [26]. Copyright 2007 American Chemical Society.

When $H_2^{18}O$ was used in the FTIR measurements, the observed isotope shift of the peak was small. Then, they suggested that one of the O atoms of the Ti-OOH group originated from O of the surface bridging O of the crystal [27]. Salvador [28] reported that the photoinduced holes are trapped at the bridging O of the surface, as stated above. However, in his proposed mechanism, a pair of •OH radicals are produced near the trapped hole, and they form the Ti-OO-Ti surface structure [28].

As shown in Figure 3, Nakamura et al. assigned the 838 cm$^{-1}$ peak in the FTIR spectra to the OO vibration of the Ti-OOH group and the peak at 812 cm$^{-1}$ to the Ti-OO-Ti group. We also measured the FTIR spectra of the adsorbed $H_2O_2$ on rutile $TiO_2$ powders and the reaction intermediates under the UV irradiation [31]. However, no absorption peaks at around 800 cm$^{-1}$ could be observed, but the peaks around 1000 cm$^{-1}$ and 934 cm$^{-1}$, which could be assigned to Ti-OOH and Ti-OO-Ti, respectively, were observed [31]. This assignment is supported by the result of the DFT calculation described later, in which the O-O stretching mode of Ti-OOH was calculated to be 1002 cm$^{-1}$ with a large intensity. Since we did not use an electron scavenger in our FTIR experiments, $O_2$ in air was reduced, and the signals of superoxo Ti-OO$^-$ were increased. On the other hand, Nakamura et al. used $Fe^{3+}$ ions as the sacrificial electron scavenger. Since the FTIR signal of Fe(IV)=O appears at 830–840 cm$^{-1}$ as reported in the literature [32], the signal at 838 cm$^{-1}$ probably originates from an Fe ion compound and should not be attributable to Ti-OOH. Due to the uncertainty of the assignments of FTIR spectra, it is not clearly concluded that the bridging O atom is involved in the Ti-OO-Ti structure.

### 3.1.4. Experimentally Suggested Mechanism for Rutile and Anatase

We have experimentally investigated the reactive oxygen species, such as •OH, $H_2O_2$, and •$O_2^-$, which are generated from $H_2O$ by $TiO_2$ photocatalysis and photoelectrodes [33]. Nakabayashi and Nosaka examined the facet dependence of water oxidation at anodically polarized $TiO_2$ single-crystal photoelectrodes [34]. Though the Faraday efficiencies of the oxygen evolution were almost 100%, the intrinsic photocurrent was increased in the order of (100) < (110) < (001). On the other hand, the formation of •OH radicals simultaneously measured was reverse order; that is, the Faraday efficiencies were 0.59, 0.23, and 0.13%. Only for the crystal of (100) facet does the photocurrent decrease with the irradiation time, but it could be recovered by depolarization of the electrode. This observation showed that the surface structure changed to inhibit the $O_2$ production at the surface of (100) single crystals. Furthermore, by the addition of $H_2O_2$, the formation of •OH radical was increased for the (100) and (110) crystals [35], indicating that $H_2O_2$ was not formed from •OH radical but $H_2O_2$ produced •OH radical at the $TiO_2$ photo-anodes. Namely, the Ti-OO-Ti structure promotes the •OH radical generation. Thus, •OH radical could be formed in the process of the oxidation of Ti-OO-Ti species, but •OH radical is not the precursor of the Ti-OO-Ti formation. This conclusion contradicts the reaction mechanism suggested by Salvador [28] in which a pair of •OH radicals form the Ti-OO-Ti structure.

When the rutile surface is the ideal crystal structure without reorganization, the Ti-OO-Ti structure of each crystal surface can be illustrated as in Figure 4. The difference in the $O_2$ production efficiency could be explained by the Ti-OO-Ti configurations for three kinds of crystal surfaces [34]. When the two Ti ions forming bridged O could not alter their position, the formation of these structures seems difficult. Furthermore, since the dihedral angle of $H_2O_2$ is 93.4°, the Ti-OO-Ti structure cannot lay on a plane, but the Ti-O-Ti can. Thus, the formation of Ti-OO-Ti at the rutile surfaces of (100), (110), and (001) seems to have some difficulty in configuring the structure.

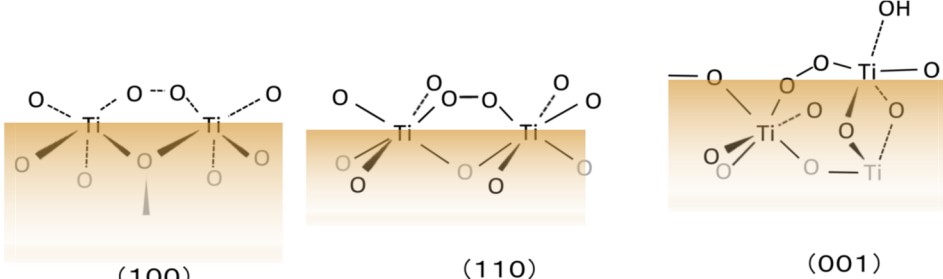

**Figure 4.** Local configuration of peroxo, Ti-OO-Ti, at distinct rutile $TiO_2$ surfaces. Adapted with permission from Ref. [34]. Copyright 2013 American Chemical Society.

In order to discuss a rational reaction model for the oxidation of water at the surface of TiO$_2$ crystal, we simply suggested that the oxidation and reduction sites are assigned to bridging OH and terminal Ti-OH, respectively [33], based on the fact that the surface Ti is positively charged while the surface O$_{br}$ is negatively charged, as expected from the following equations of ionization equilibrium in aqueous solution.

$$\equiv \text{Ti-OH} \rightleftarrows \equiv \text{Ti}^+ + \text{OH}^- \tag{9}$$

$$[\text{Ti-O}^- \text{-Ti}] + \text{H}^+ \rightleftarrows [\text{Ti-OH-Ti}] \tag{10}$$

Figure 5 shows the plausible water oxidation processes at the bridging O site of the anatase and rutile TiO$_2$ surfaces [31]. A photoinduced hole attacks the bridging O atom at first, followed by the attack of water resulting in the formation of a pair of Ti-O$\bullet$ and Ti-OH [29]. As shown in Figure 5, at the anatase surface, since the Ti-Ti distance is too large to form peroxo Ti-OO-Ti structure, a surface-trapped hole should be isolated, and it is useful to oxidize organic molecules RH. A certain fraction of the trapped holes desorbs as $\bullet$OH radicals into the solution [36,37]. On the other hand, at the surface of rutile, when a second hole is generated in the particle, it migrates to combine with the existing hole to form bridging peroxo species at the surface [29,38]. The distance between Ti atoms at the rutile surface (2.96 Å) could be shorter than that of anatase (3.79 Å) and be favorable to forming the Ti-OO-Ti structure [31,39]. Thus, rutile crystal is relatively more active for the water oxidation to O$_2$ evolution against anatase crystal [33,36]. This explanation by the surface Ti-Ti distance for the difference in the water oxidation between rutile and anatase was supported by a theoretical calculation in a recent report [40].

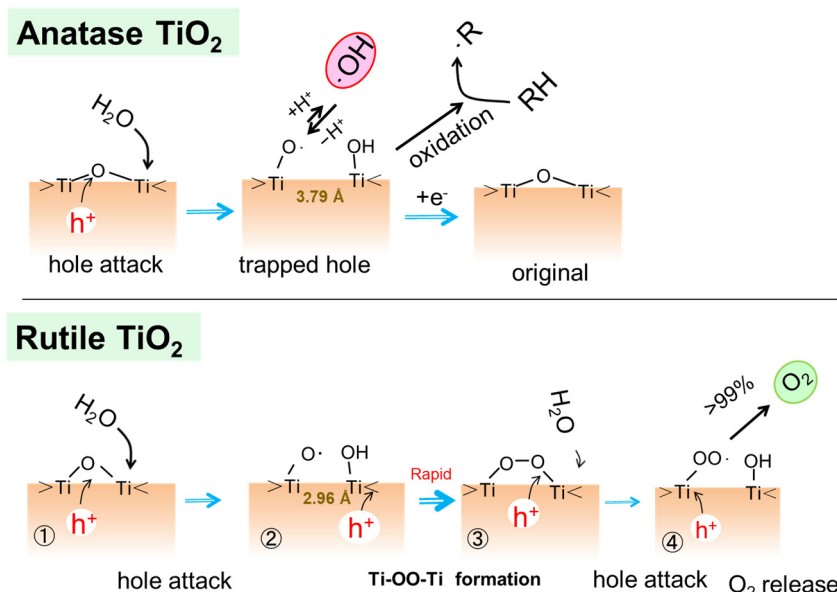

**Figure 5.** Plausible photocatalytic oxidation paths at the surface of TiO$_2$ of anatase and rutile polymorphs. Reproduced from Ref. [31] with permission from the PCCP Owner Societies.

When the peroxo Ti-OO-Ti is further oxidized with a third hole, superoxo structure Ti-OO$\bullet$ is formed [26,29]. This radical could release superoxide radical $\bullet$O$_2^-$ or be further oxidized to release O$_2$ and then return to the bridging O. The generation of $\bullet$O$_2^-$ on the addition of H$_2$O$_2$ was supported by the experiment for rutile powder [31]. On the other hand, when the O-O bond in the Ti-OO-Ti structure is cleaved by a third hole, Ti-O$\bullet$ is formed, and it could return to the Ti-OO-Ti by releasing $\bullet$OH, similar to the case of the trapped hole for anatase. This mechanism of the $\bullet$OH generation explains the experimental result that in the presence of H$_2$O$_2$, the generation of $\bullet$OH increased in the photoelectrochemical experiments [35].

The surface structure of the Ti-O-Ti group is fixed due to the formation of a four-membered ring with Ti and O atoms. Therefore, Nakamura et al. suggested that the photoinduced hole could not be trapped at the terrace, and the 3-coordinated O ($O_{3c}$) at the step or corner of the crystal surface may be the location for the hole trapping. The above discussion is based on the observation by UPS measurement, in which the energy level of the O atom in terminal Ti-OH was deeper than that of the valence band top [28]. However, it was confirmed only for the (100) surface. They did not discuss the energy level for the other crystal surface. The •OH radical may be generated at the Ti-OH part [25] because the •OH generated by ionization radiation has the redox potential the same as the surface trapped holes [41], and then it is suggested to coordinate to a surface Ti ion [33].

From a kinetic analysis of TiO$_2$ photoelectrodes, Kafizas et al. [42] concluded that the rate-limiting step of the catalytic cycle changes with pH. In alkaline pH, three accumulated holes are required in the rate-limiting step, while two accumulated holes are required in neutral and acidic pH. Since the rate of water oxidation is similar over the wide range of pH, a hydroxy nucleophilic attack is most probably not included in the rate-limiting step. Then, the nucleophilic attack to form Ti-O• radical may not be supported for usual rutile powders [42]. In Section 3.3, we will show how terminal OH can be oxidized by the hole at bridging O based on the theoretical calculation for a surface model molecule.

### 3.2. Theoretical Calculation Approach

In each oxidation step of OER, the electron transfer reaction is coupled with the proton transfer reaction. The effects of the TiO$_2$ surface on the electron and proton transfer processes were analyzed by a hybrid DFT method [43]. The energy level diagram for fully decoupled electron- and proton-transfer reactions were described as shown in Figure 6 [43]. Red and blue represent electronic and protonic levels, respectively. Solid lines are the levels calculated on aqueous rutile TiO$_2$, and the representative experimental levels in aqueous solution are shown by dashed lines for comparison. The catalytic effect in each step can be known by the difference between the solid line and the dashed line. The black arrow represents the dehydrogenation potential because the dehydrogenation potential is equal to the sum of the deprotonation and oxidation potentials. The adsorption on TiO$_2$ greatly reduces the first dehydrogenation potential by 0.7 V, indicating that the energy for the first step becomes lower by the adsorption [43]. However, in this energy calculation, the second step was the formation of a bi-radical, Ti-O$^{••}$. Then, the effect of TiO$_2$ adsorption was similar value for both the electron and proton transfers, and then the dehydrogenation potential was slightly increased in this calculation. Thus, it is indisputable that the key point in the OER is the surface reaction in the second step, that is, the formation of (hydro)peroxo, Ti-OOH, or Ti-OO-Ti.

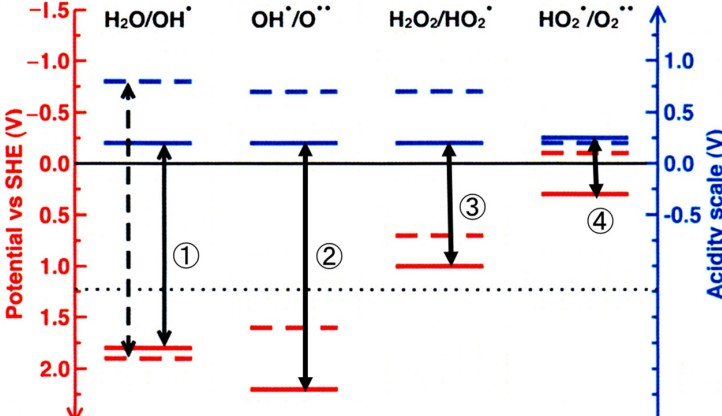

**Figure 6.** Electronic and protonic energy level diagram for the four proton-coupled electron transfer steps in water oxidation. Dashed lines are experimental data in aqueous solution, and solid lines were calculated for aqueous rutile TiO$_2$ surface. Adapted with permission from Ref. [43]. Copyright 2014 WILEY-VCH Verlag GmbH & Co. KGaA.

Di Valentin reported a study of first-principles DFT calculation based on a hybrid functional method for the hole-mediated water photo-oxidation at anatase $TiO_2$ (101) surfaces [44]. In their conclusion, a water molecule adsorbed at $Ti_{5c}$ is found to directly interact with a self-trapped hole at a bridging oxygen $O^-_{br}$ site and to transform into an ●OH radical species, as shown in Figure 7a. The ●OH radical shift the adsorption site from the $O^-_{br}$ to the $Ti_{5C}$ as shown in Figure 7b. Thus, the second ●OH radical of the next $Ti_{5C}$ site combines with the ●OH radical to form adsorbed $OOH^-$ as shown in Figure 7c. The key step of the ●OH formation was very similar to Equation (8) which was deduced from the STM observation. Thus, ●OH is formed as expressed by the following equation [44].

$$H_2O + h^+(O^-_{br}) \rightarrow \bullet OH + {}^+OH_{br} \tag{11}$$

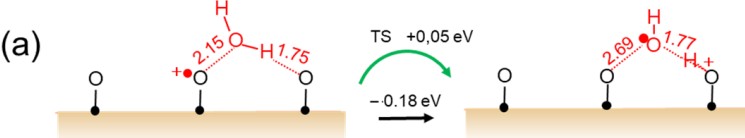

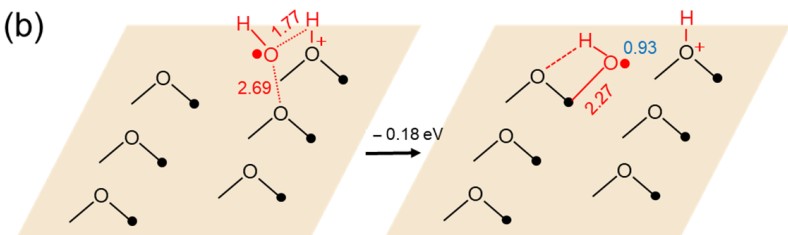

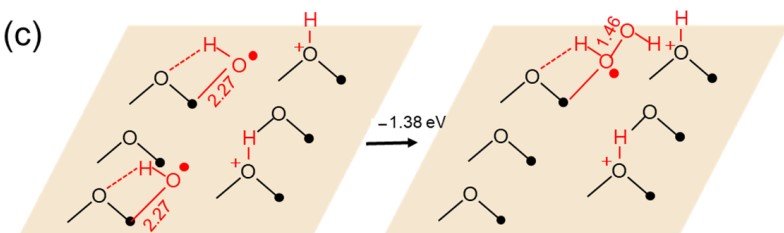

**Figure 7.** The key steps in OER at anatase $TiO_2$ (101) surface. (**a**) The proton transfer from the adsorbed water molecule to a bridging O atom leading to the formation of an ●OH radical species. (**b**) The ●OH radical transfer to the $Ti_{5c}$ site. (**c**) The coupling of two ●OH radical species to form one $H_2O_2$ molecule. Small balls represent surface Ti ions. Distance (in the unit of Å) and electron spin density were numerically shown. Adapted with permission from Ref. [44]. Copyright 2016, IOP Publishing.

Li and Selloni [45] used hybrid density functional-based energetic calculations and first-principle molecular dynamics simulations to investigate the pathway and kinetics of the OER on the anatase $TiO_2$ (101) surface in a water environment. The snapshots of the oxidation process are shown in Figure 8. They assumed direct oxidation at the $H_2O$ coordinated to Ti ions, which was not supported due to the positive charging effect as described above. Though the first oxidation site is $O_{br}^-$, the formed ●OH radical is probably stabilized at the surface Ti ion. Therefore, the resultant structure of the first step is the same. Oxidation of Ti–OH gives rise to a second stable intermediate, a surface bridging peroxo dimer, $(O_2^{2-})_{br}$, composed of one water and one surface lattice oxygen atom, consistent with the surface peroxo intermediates reported by Nakamura et al. from the FTIR measurements on rutile [26,27]. The calculations predicted that molecular oxygen

evolves directly from $(O_2^{2-})_{br}$ through a concerted two-electron transfer, thus leading to oxygen exchange between $TiO_2$ and the adsorbed species. They also described that the nuclear attack mechanism proposed by Nakamura et al. [26,27] for rutile using the configurations with the hole at $O_{br}$ was found to be energetically unfavorable on the anatase (101) surface. On the rutile (110) surface, O add atoms have a terminal geometry so that Ti-OOH can readily form via the nucleophilic attack of a water molecule. Thus, they suggested that different OER pathways are likely to be operative on the two main $TiO_2$ polymorphs and then observed lower OER activity of anatase relative to the rutile [45].

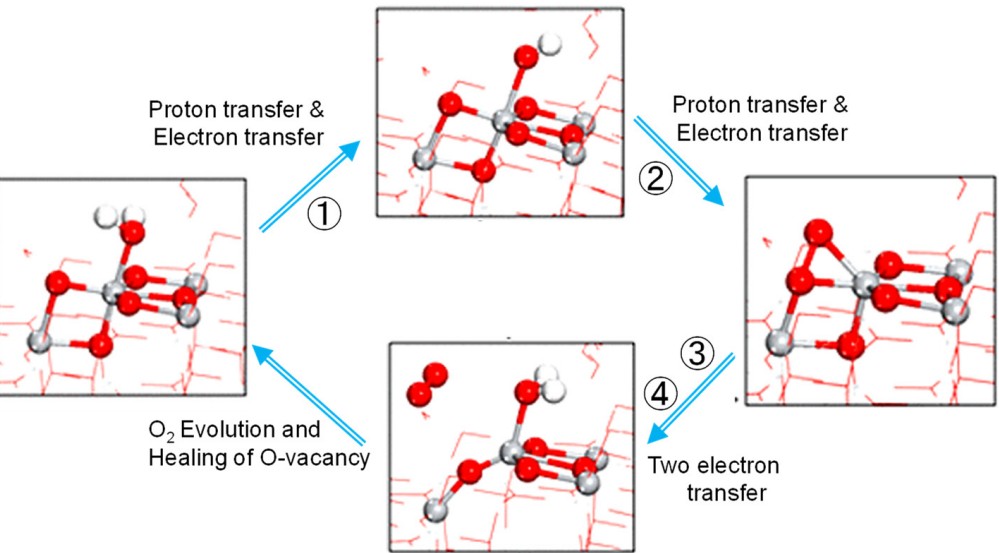

**Figure 8.** Water oxidation snapshots on anatase (101) surface deduced from a theoretical calculation. $Ti-OH_2 \rightarrow Ti-\bullet OH \rightarrow (O_2^{2-})_{br} \rightarrow O_2$. Adapted with permission from Ref. [45]. Copyright 2016 American Chemical Society.

Wang and co-workers [46] developed a first-principles multi-point averaging molecular dynamics method to extensively simulate the OER in the presence of liquid water on rutile $TiO_2$ (110). They first calculated the dissociation of $H_2O$ adsorbed on the surface and found that the hole trapping at $OH_t^-$ is 0.24 eV more favorable than that observed for $O_{br}^{2-}$. Additionally, compared to the situation in the gas phase, the absolute values of the hole-trapping capacity of surface hole traps were remarkably reduced by 0.6 eV in the presence of liquid water. The formation of surface radicals $\bullet OH_t$ and $\bullet O_{br}^-$ and further oxidation pathways on the Ti row and $O_{br}^{2-}$ row on the rutile (110) facet were illustrated in Figure 9. The experimental observation that the surface roughening on rutile(110) was significantly suppressed at pH 13 [26] was explained by the result that the lack of $\bullet OH_t$ led to the decrease in the rate of pathway II and hence the reduction of surface roughening. They simulated O-O stretching frequencies for three peaks in the literature 840 cm$^{-1}$, 816 cm$^{-1}$, and 928 cm$^{-1}$, to the corresponding three species $(O_t-O_t)^{2-}$, $(O_{br}-OH)^-$, and $(O_t-OH)^-$, respectively. These assignments do not agree with ours, as stated above. Their calculation results may explain the experimental results for rutile (110) crystals. However, the reaction mechanism in Figure 9 is not supported by the results in the experimental approach with ESR, NMR, and STM stated above. Furthermore, the (110) facet is not the only surface of the rutile $TiO_2$ powder. Therefore, it may need careful consideration to adopt this water photo-oxidation mechanism for powder photocatalysts.

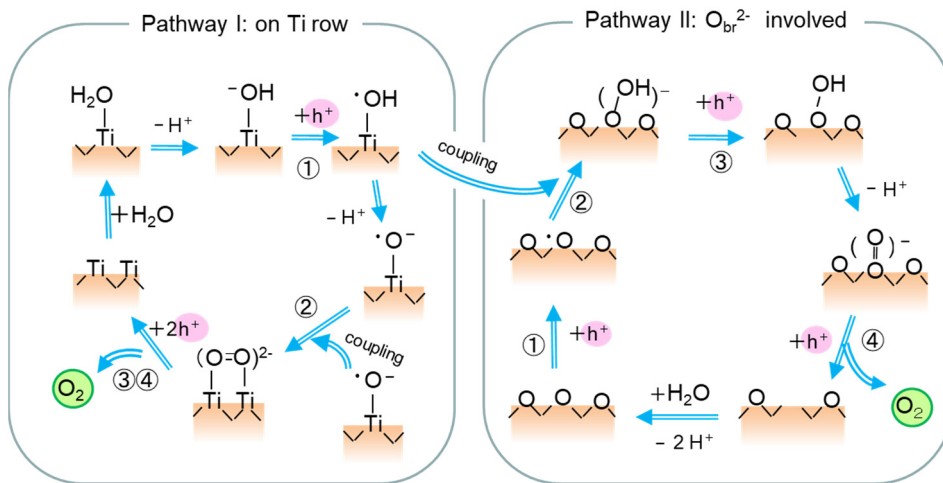

**Figure 9.** Dual pathways mechanism for photocatalytic OER at the surface of rutile $TiO_2$ (110) facet proposed from the multi-point averaging molecular dynamics approach [46].

### 3.3. Molecular Mechanism at Rutile (011) Surface

As stated above, photocatalytic water oxidation over rutile $TiO_2$ is superior to that over anatase $TiO_2$ [33,35]. For the photocatalytic reactions over rutile $TiO_2$ particles, it was reported that the oxidation occurred at the (011) surface and the reduction at the (110) surface from the photodeposition experiments for Pt and $PbO_2$ [47]. However, the research on the reactions at the rutile (011) surface was very few [48], and this surface was not used for the theoretical calculation as far as the author knows. According to the STM observation of rutile (011) surface in a vacuum, O atoms are adsorbed by forming a (2 × 1) surface structure [25]. However, since the actual photocatalytic reactions are performed in water, the adsorption of water molecules probably sustains the (011) surface structure of a crystal. Then, the water oxidation mechanism was explored by a DFT calculation method with a model molecule for the rutile (011) surface [49].

The model molecule $Ti_5O_{19}H_{16}$ used was shown in Figure 10, which was constructed from a rutile crystal model in software named Avogadro. As shown in Figure 10, the model molecule consists of an eight-membered ring of the Ti-O- sequence. Among the four Ti atoms marked A, B, C, and D, two bridging O atoms located among $Ti_B$-$Ti_C$-$Ti_D$ sit upward of the surface, while the other two sit under the surface. In order to prepare a stable molecule, another Ti atom was added at the bottom. When axial O atoms were added on $Ti_A$ and $Ti_B$, as shown by two blue balls in Figure 10, the separation between the two axial O toms was only 2.0 Å and built a Ti-OO-Ti structure with the dihedral angle of 76° before the structural optimization in the calculation. The theoretical calculation was performed with the DFT method of B3LYP/6-31G(d) with the option of partial geometry optimization in the Gaussian03W program [49]. Though the 6-31G(d) basis set was not recommended in the recent review paper [50], we employed it because it is the most commonly used basis set [50].

The energy levels for many plausible structures of intermediate spices were calculated and some of which are shown in Figure 11. In the first step of the oxidation, the electron spin distribution of the added hole was confirmed to locate at the bridging O between $Ti_B$ and $Ti_C$. When an $H_2O$ molecule was coordinated to the $Ti_A$ ion, electron spin appeared at O of the coordinated $H_2O$. Simultaneously, its H atom was coordinated with the facing $O_{br}$ atom. The coordination of •OH radical to Ti atom causes the ionization of •OH radical to form •$O^-$ radical as discussed in our previous report [33]. Thus, it was confirmed that Ti-O• radical, or trapped hole, could be formed at the position of terminal Ti-OH by the trapped hole at $O_{br}$, in which the deference from the mechanism in Figure 3 is that the Ti-O-Ti structure is not broken.

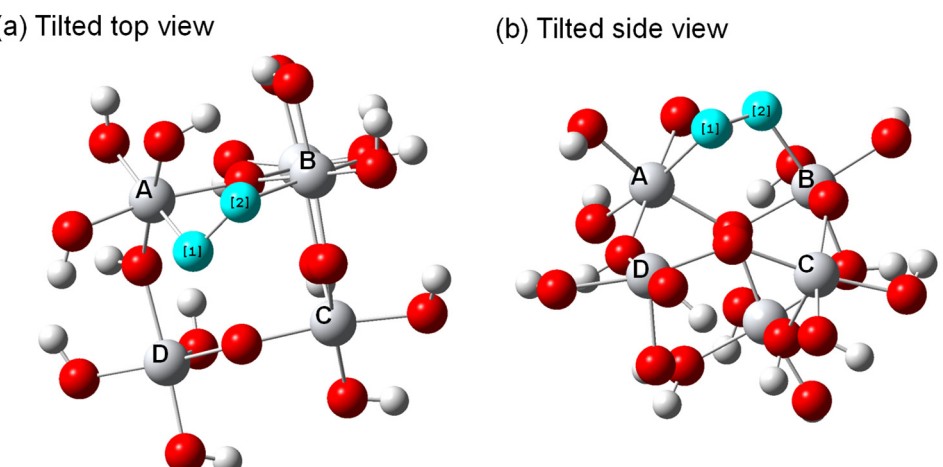

**Figure 10.** The structure of the model molecule $Ti_5O_{19}H_{16}$ for rutile $TiO_2$ (011) surface with the addition of two axial oxygens (blue color) forming stable surface Ti-OO-Ti structure. (**a**) Tilted top view and (**b**) tilted side view. Adapted from Ref. [49].

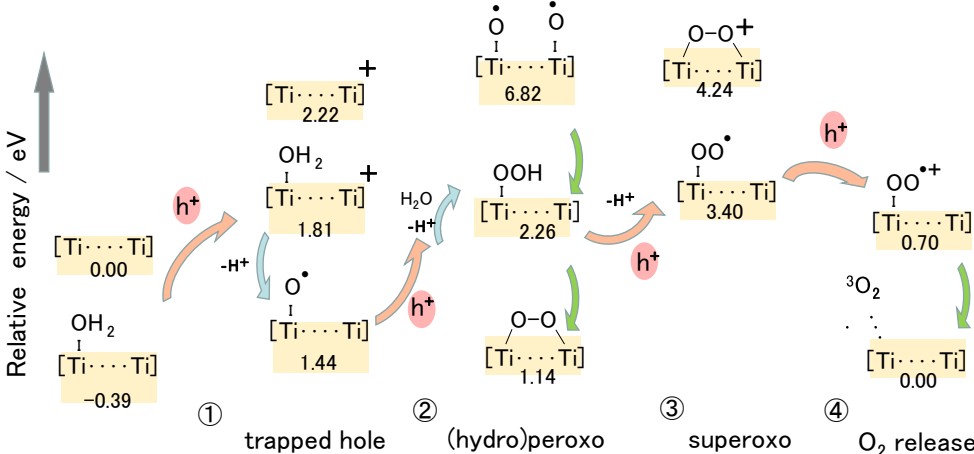

**Figure 11.** Difference in the total energy of the intermediates in the water oxidation for the model molecule of rutile $TiO_2$ (011) surface. [Ti . . . Ti] stands for $Ti_A$ and $Ti_B$ ions at the surface. Adapted from Ref. [49].

At the second step of the water oxidation, due to the symmetrical surface structure, Ti-O• radical may be formed at the $Ti_B$ by transferring the hole from $O_{br}$ of $Ti_C$-O-$Ti_D$. However, the energy of the structure of the bi-radical was too high to be considered, as shown in Figure 11. Actually, in the DFT calculation, the second hole is located at the O atom of the Ti-O• radical. Thus, the oxidation of $H_2O$ coordinated to $O_{br}$ of $Ti_B$-O-$Ti_C$ is reasonably oxidized by the $Ti_A$-O• radical to form a hydroperoxo ($Ti_A$-OOH) structure. By releasing the proton from Ti-OOH and coordinating with the $Ti_B$ site, the peroxo (Ti-O-O-Ti) structure could be formed with a low energy level, as shown in Figure 11.

When the next hole comes at the third step, a stable Ti-OO-Ti structure was not necessary as an intermediate to form superoxo (Ti-OO•) radical since the energy of Ti-OO• radical was lower than the oxidized Ti-OO-Ti structure, as shown in Figure 11. In other words, oxidation of the Ti-OO-Ti site is not easy. Since the Ti-OO• radical is equivalent to the superoxide •$O_2^-$ radical coordinated to the surface Ti ion, at the fourth step of water oxidation, the Ti-OO• radical could easily release a triplet $O_2$ molecule to back to the original surface structure.

In these four oxidation steps in Figure 11, the rate-determining process is the first two steps because of the highest energy difference of 2.2 eV. Since the three Ti atoms ($Ti_A$, $Ti_B$, $Ti_C$) do not locate linearly, it is not easy to draw an image in the 2D illustration. However,

the detail of these key steps can be shown in Figure 12. In the first step, a photoinduced hole reaches the surface $O_{br}$ to form $Ti_B$-$O^{\bullet}$-$Ti_C$, followed by the oxidation of the water molecule adsorbed at this $O_{br}$ to form $\bullet OH$ radical. This $\bullet OH$ radical rapidly coordinates with the $Ti_A$ ion which is facing the $O_{br}$ of $Ti_B$-O-$Ti_C$. In the second step, when a photoinduced hole reaches the $Ti_A$-O$\bullet$, a water molecule coordinated to the previous $O_{br}$ is oxidized at the $\bullet O$ radical on $Ti_A$ to form a hydroperoxo Ti-OOH structure. The reaction mechanism of these first two steps is in harmony with the reported results stated above. Those are the soli-state NMR observation by Liu et al. [23], the STM observation by Tan et al. [24], and the DFT calculation by Di Valentin [44] described above. Since the $\bullet OH$ radical does not coordinate with the Ti ion, which has bonded the oxidized $O_{br}$, the reaction mechanism is different from that proposed by Nakamura et al. in Figure 3 [26]. Thus, it was clarified that the surface arrangement of Ti and O atoms at the rutile (011) surface is favorable to efficiently progressing proton-coupled electron transfer reaction to generate molecular oxygen.

(a) 1ˢᵗ step                                                     (b) 2ⁿᵈ step

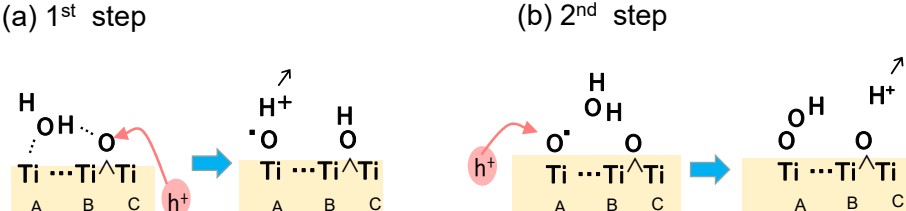

**Figure 12.** Key steps in water oxidation deduced from the DFT calculation for the model molecule of rutile $TiO_2$ (011) surface.

## 4. Conclusions

Fifty years ago, the stable characteristics of $TiO_2$ as a photosensitizer were discovered. Starting from this discovery, the annual number of scientific reports on photocatalysis increased exponentially, up to tens of thousands 20 years ago, and most of these reports treated $TiO_2$ as photocatalysts [1]. The photocatalysts of the reports in the 1990s were mainly used for environmental cleaning or disinfection, which could be easily used as practical applications. The fatal disadvantage of photocatalytic water oxidation with $TiO_2$ is its band-gap energy which determines the wavelength of the photon absorption. Only a few % of the UV region in the solar light spectrum can be used. However, from the scientific point of view, the reaction mechanism of OER over $TiO_2$ has been under discussion up to now. Though $TiO_2$ has not been employed as a co-catalyst of visible-light-responsive semiconductor photocatalysts, the developments in the research techniques for $TiO_2$ photo-oxidation are useful for improving the photocatalytic activity of other photocatalysts and co-catalysts.

There are two methods to investigate the reaction mechanism, namely the experimental approach and the theoretical calculation approach. In the experimental approach, detection methods to investigate reaction mechanisms may be limited in some techniques. However, the observed results could be concrete as the distinct experimental method. Sometimes, no results could be obtained in the individual experiments. On the other hand, in the theoretical calculation, the results could always be obtained. Thus, different calculation methods may lead to different conclusions. Then, the theoretical calculation methods should be tested by comparing them with the experimental results. When the results in theoretical calculation accord with the experimental results, the calculated results become useful to visualize the reaction mechanism. Then, the assessment of the experimental results seems important for the theoretical calculation to be utilized.

In the present report, seven kinds of reaction mechanisms of Figures 2, 3, 5, 7–9 and 11 were described for OER at the $TiO_2$ surface in water. The well-known FTIR experiment by Nakamura et al. [27] contradicted our experimental results [31], as described above. Therefore, their conclusion that surface lattice O atom was involved in the produced $O_2$ may not be real, suggesting that the reaction mechanisms in Figures 2, 3, 5 and 9 become doubtful, at least for rutile $TiO_2$. Another problem in the trustability of the theoretical

calculations in the literature was that the facets of $TiO_2$ were limited to anatase (101) and rutile (110). The extensive theoretical calculation for other facets is needed to describe the concrete OER mechanism over $TiO_2$.

**Funding:** This research received no funding.

**Conflicts of Interest:** The author declares no conflict of interest.

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
