# Peer review of "Water Photo-Oxidation over TiO2—History and Reaction Mechanism"

_catalysts, doi:10.3390/catal12121557_

Round 1

Reviewer 1 Report

This manuscript is a complete review of photocatalytic water splitting using TiO2 as catalyst. It is a very useful document for researchers starting working on this subject including some of the history of this process, systems used and a complete explanation of the reaction mechanism. I have not found mistakes neither in the text nor in the figures and, in my opinion, this manuscript can be published in Catalysts only adding a Conclusions section at the end of the text.

Author Response

According to your suggestion, the section of “Conclusion” was added in the revision.  

Reviewer 2 Report

Overall, the article does fulfill the expectations one gets from this title. The goal of the paper is to focus on the history and reaction mechanism of photocatalytic water oxidation over titanium. Within this context, I would like to give some advice. Below are some aspects I have noted and which I would like to see addressed in a suitably revised version of this review article. It deserves publication considering some minor comments.

1.       As a review paper, it lacks the perspective section. After reading a review of the history of water photooxidation over TiO2, the audience may wonder what is next in this field. If possible, I seriously invited the respected author to add a short perspective at the end of this review, to point out what worth people focus on in this field.

2.       Although the topic is much more traditional, a review should also cover some cutting-edge content, and some recently-published top papers can be considered to be cited if necessary, such as Nature Catalysis, 2018, 1(4): 291-299 and Nature Chemistry, 2020, 12(1): 82-89. For the experimental part, more in situ/operando studies can be included since the review focus much on mechanism discussion. And for the computational part, DFT alone may not necessarily reveal the dynamic mechanism. Some other modeling methods like molecule dynamics can be involved.

3.       The author seems involved in their research studies in this review paper. I think it is OK if necessary. However, the introduction part should reduce the ratio of inducing the research parts, as it may a little bit confusing about what is this review paper targeting. And in the introduction, the author said, “Since the OER activity of rutile TiO2 is higher than that of anatase, and rutile (011) surface has been assigned to the oxidation facet, we performed a DFT calculation for a (011) surface model molecule.” It is not a solid statement why the author performs this study as it is not pointed out the scientific question.

4.       B3LYP/6-31G(d) is quite old-fashion, some more reliable methods should be considered. For more detail can refer to Angew. Chem. 2022, e202205735. As the process may involve photoexcitation, a Time-dependent DFT calculation can be also considered if possible.

Author Response

  1. I added some perspective comments in the section of “Conclusion” added in the revised manuscript.
  2. The paper of “Nature Catalysis, 2018,” gives an important information about water photooxidation over rutile (110) facet. I added a paragraph and a new figure to describe the research in this paper. On the other hand, in “Nature Chemistry, 2020,” the water oxidation over Fe2O3 was mainly described. So, this paper was not included in the revised manuscript.
  3. This statement in “Introduction” was deleted.
  4. We added ” Angew. Chem. 2022,” as Ref [50]. According to this paper, we added the sentence that “Though the 6-31G(d) basis set was not recommended in the recent review paper [50], we employed it because it is the most commonly used basis set [50].” In the revised manuscript.

Reviewer 3 Report

The manuscript surveyed and summarized the development in the field of TiO2-based photocatalytic water oxidation reaction, including experimental approaches like ESR, NMR, STM, and also theoretical approaches discussing the reaction mechanisms. This is an excellent contribution to this research field and the manuscript is well-written, easy to follow. Really enjoy reading this manuscript and it should be acceptable for publication.

Very Minor issues:

1) Full name of the technique should be provided, like ESR, NMR.

2) Figure 2 is partially not very clear and not easy to read. 

Author Response

1)Full name of the technique was provided in the revised manuscript.

2)Figure 2 was corrected to clarify the letters.

Reviewer 4 Report

1. The first abbreviation should be explained clearly.

2. The typesetting errors should be revised clearly, such as

•OH as •OH-

3. Some references lack of the important information, such as the year, volume, and pages. Please check it carefully. 4. Some recent related work is missing. 5. Review all graphics, subtitles are small, ariel, no pattern. This seems irrelevant but it organizes the work for the reader.

6. In the introduction, the research progress of TiO2 or carbonized TiO2 should be provided instead of MOF, such as CrystEngComm, 2022, 24, 6933–6943; J. Alloy. Compd, 2022, 897, 163178 and Catal. Sci. Technol., 2021, 11, 3946–398.

7. Some updated refs on the Photocatalytic splitting of water could be compared, such as Appl. Surf. Sci., 2020, 515, 146038; J. Catal.,2021, 394, 397-405. Chem. Eng. J. 2022, 431, 134072; Appl. Catal. B: Environ. 2022, 306, 121095.

8. The section related to the title topic lacks a discussion concerning the pros and cons of TiO2. Authors may look into it. 9. Conclusion and challenges section seems to be too shorter. Please try to illustrate it.  

Author Response

  1. The abbreviation was not used as keywords in the revised manuscript.
  2. I could not find a typesetting error of OH-.

   I think that •OH is correct typesetting.

  1. In the Reference section, I could not find a lack of word. In the Figure captions, word of “Ref.” was added to the reference number.
  2. Acceding to the other reviewer, the literature of “Nature Catalysis 2018” was added in the references.
  3. We revised Figure 2 as the suggestion of other reviewer.
  4. Suggested references deal with MOF for HER with sacrificial electron donor. As for MOF photocatalysts, the present author thinks that it is not suitable for multi electron reaction such as OER. To describe this idea, we added following sentences in Introduction.

Since these electron transfer reaction originated from the absorption of one photon for each step, semiconducting properties of metal oxide solid materials are useful to promote the reaction at a reaction site. Thus, metal complexes and metal organic framework (MOF) are almost certainly unsuitable as the photocatalysts for OER [2].

7 . The following literatures you suggested were checked

“Appl. Surf. Sci., 2020.” ; It escribed methanation of CO2. TiO2 was not used

“J. Catal.,2021”ï¼› It described methanation of CO2. TiO2 was not used

“Chem. Eng. J. 2022” ; It described OER but it was not photooxidation. RuO2 was used as the OER catalyst.

“Appl. Catal. B: Environ. 2022”;  It does not use TiO2. Described only HER

Thus, the literature you cited are not for water photooxidation and the out of scope of this review paper.

8,9 We added conclusion section to describe “pros and cons”.